# The Role of Skin Tests in the Prevention and Diagnosis of Hypersensitivity Reactions to Platinum Agents in Gynecological Cancer: A Single-Center Italian Retrospective Study

**DOI:** 10.3390/cancers13215468

**Published:** 2021-10-30

**Authors:** Ilaria Puxeddu, Fiorella Petrelli, Maria Elena Guerrieri, Stefania Cosio, Isabella Del Corso, Valeria Rocchi, Maria Laura Manca, Paola Migliorini, Angiolo Gadducci

**Affiliations:** 1Immunology and Allergology Unit, Department of Clinical and Experimental Medicine, University of Pisa, 56127 Pisa, Italy; ilaria.puxeddu@unipi.it (I.P.); fiorella.petrelli@phd.unipi.it (F.P.); i.delcorso@ao-pisa.toscana.it (I.D.C.); valeriarocchi000@gmail.com (V.R.); paola.migliorini@unipi.it (P.M.); 2Division of Gynecology and Obstetrics, Department of Clinical and Experimental Medicine, University of Pisa, 56127 Pisa, Italy; mariaelenaguerrieri@gmail.com (M.E.G.); s.cosio@ao-pisa.toscana.it (S.C.); 3Internal Medicine Unit, Department of Clinical and Experimental Medicine, University of Pisa, 56127 Pisa, Italy; l.manca@med.unipi.it

**Keywords:** gynecological cancer, carboplatin, cisplatin, hypersensitivity reactions

## Abstract

**Simple Summary:**

The development of hypersensitivity reactions to platinum agents in patients with gynecological cancers limits the use of platinum re-treatment for recurrent disease. In those patients who developed a hypersensitivity reactions during treatment with platinum agents it could be safer to undergo allergy diagnosis. This approach includes the *in-vivo* skin tests before re-exposure to the platinum agent, especially in those patients who have to undergo a 2nd or 3rd line therapy. In our experience, skin test for platinum agents resulted in a simple and sensitive tool for the diagnosis and prevention of hypersensitivity reactions to platinum agents. In addition, this approach identified a sub-group of patients that became sensitized to the platinum agent during the previous platinum-based therapy.

**Abstract:**

Background: Hypersensitivity reactions (HSR)s to platinum agents are increasing in frequency, due to their extensive use and repeated exposures in patients with increased life expectancy. The aims of our study are to analyze the frequency of both type I and type IV HSRs in patients with gynecological cancer treated with (CBDCA) carboplatin and/or (CDDP) cisplatin, to evaluate the role of skin tests in the diagnosis and prevention of HSRs. Methods: From 2011 to 2018, we evaluated 124 consecutive female patients previously treated with CBDCA and/or CDDP for gynecological cancer. All patients, including those with and without HSR to previous platinum-based therapy, underwent *in-vivo* skin tests for platinum agents before starting the second or more therapeutic lines. To reduce the risk of false negative results, patients with a negative skin test at the first evaluation were re-tested after 3 weeks from the platinum re-exposure. Results: Among the 124 patients evaluated, 58 (47%) experienced HSRs to at least one platinum agent: 35% were to CBDCA, 5% to CDDP, 7% to both. Fifty-six of the 58 HSRs were classified as immediate and two delayed. Skin tests confirmed an IgE-dependent mechanism in 67% of patients with immediate-HSRs to CBDCA and identified a cross-reactivity between platinum agents in 18% of patients. Moreover, among those who had never developed an HSRs during platinum-based therapy, *in-vivo* skin tests identified 12% of sensitized patients. Conclusions: On the basis of our findings, skin test for platinum agents is a simple and sensitive tool for the diagnosis and prevention of HSRs to CBDCA and/or CDDP and can be useful for detecting possible cross-reactivity among platinum agents.

## 1. Introduction

Platinum-based chemotherapy is widely used in the treatment of primary and recurrent gynecologic malignancies. Hypersensitivity reactions (HSRs) to platinum, including anaphylaxis, are increasing in frequency due to their extensive use and repeated exposures in patients with increased life expectancy. After a reaction to carboplatin or cisplatin has occurred, it is difficult to use a platinum agent, despite patients having platinum –sensitive disease. In particular, HSRs to carboplatin (CBDCA) affect almost 2% of the general oncologic population and 16% of patients with ovarian cancer [1,2,3]. Furthermore, increasing the cycles of platinum-based chemotherapy enhances the probability to develop HSRs to platinum agents, which exceeds 27% in patients who received more than seven cycles of treatment [4]. In these patients, the administration of cisplatin (CDDP) or oxaliplatin may be an option, even if the possibility of having cross-reactivity should be considered [5,6].

Several studies have tried to identify risk factors for developing HSRs to platinum agents. A previous exposure to these agents seems to be the primary risk factor, followed by total lifetime exposure, a long interval from the last platinum exposure, atopic status, and previous systemic allergic reactions to other platinum agents [2,7,8]. Deleterious *BRCA 1/2* mutations have recently been identified as independent risk factors for HSRs to platinum [9]. 

The HSRs to platinum agents can be immediate (type I) or delayed (type IV). Whereas the immediate-HSRs are supported by the rapid onset of symptoms and an IgE-mediated mechanism, the delayed-HSRs may occur several hours after platinum administration and the mechanism involved is predominantly T-cell mediated [5,10]. Approximately 50% of CBDCA-induced immediate-HSRs are associated with moderately severe symptoms, consisting of diffuse erythroderma, wheezing, facial swelling, gastrointestinal symptoms (nausea and/or vomiting diarrhea), or anaphylaxis (dyspnea and/or hypotension) [4,11,12].

*In-vivo* skin tests, specific IgE and basophil activation tests (BAT) are all possible tools to identify an IgE-dependent mechanism in platinum-induced HSRs, although the *in-vivo* skin tests are the only tools currently used in the clinical practice [11,13,14,15]. In addition, skin tests are able to detect both a type I and a type IV reaction, performing both immediate and delayed readings. 

Furthermore, it has been shown that performing skin tests for CBDCA before starting a new line therapy for recurrent disease may identify patients that have been sensitized to platinum agents during previous cycles of therapy [16,17,18]. Overall, the sensitivity of the skin tests for the diagnosis of immediate-HSRs to CBDCA is approximately 80%, but it may be less than 36% when the test is performed after more than 6 months from the reaction [13,14,17,18,19]. In addition, the negative predictive value of the *in-vivo* skin tests ranges between 99% and 92% in those patients who had received at least 6 previous cycles [11,13,16]. As one of the primary barriers in the use of CBDCA or other platinum agents is the development of HSRs, it is important to carry out an allergy work-up for preventing severe reactions and planning alternative therapies such as switching to another platinum agent or performing desensitization. The aims of our study are to analyze the frequency of HSRs in patients with gynecological cancer treated with CBDCA and/or CDDP and to evaluate the role of skin tests in the diagnosis and prevention of HSRs.

## 2. Materials and Methods

### 2.1. Patients 

From 2011 to 2018 we evaluated 124 consecutive female patients who received salvage platinum-based treatment for recurrent gynecological cancer after a prior treatment with CDDP and/or CBDCA at the Azienda Ospedaliera Universitaria Pisana (AOUP, Pisa, Italy), without exclusion criteria. Demographic and clinical characteristics of the patients were collected during the first allergy evaluation, including history of atopy, any HSRs during previous platinum-based therapy, and previous systemic allergic reactions to other drugs and/or contrast media. All this information was also collected during the following visits. Information regarding total cycles and lines of therapy was collected. In those patients who previously experienced HSRs to platinum agents, the reaction was classified on the basis of time of onset (immediate or delayed) and severity (grades 1–5) [20]. Furthermore, 47 patients with ovarian cancer underwent genetic testing for detection of *BRCA 1/2* mutations [21].

### 2.2. In-Vivo Skin Tests to Platinum Agents

Before platinum re-exposure for recurrent disease, *in-vivo* skin tests for both CBDCA and CDDP were performed in all the patients evaluated. In those patients with negative skin tests and their last platinum treatment more than 6 months previously, re-testing was performed after 3 weeks from platinum re-exposure. The *in-vivo* skin tests were performed using skin prick tests (SPT) at a concentration of 10 mg/mL for CBDCA and 1 mg/mL for CDDP. Intradermal tests (IDT) were performed in cases of negative SPT, using 0.02 mL of solution, containing different dilutions of CBDCA (0.01, 0.1, and 1 mg/mL) and CDDP (0.001, 0.01, and 0.1 mg/mL) [22]. SPT results were evaluated after 15 min and IDT after 20 min for the immediate-HSRs. In cases of delayed-HSRs the readings were also taken after 24, 48, and 72 h. The results of SPT and IDT were compared with the positive (histamine, Lofarma Allergeni, Milan, Italy) and negative (0.9% saline solution) controls and were assigned an arbitrary score based on the size of the wheal (>3 mm) [15]. Informed consent was obtained from each patient before performing the *in-vivo* skin tests.

### 2.3. Statistical Analysis

Continuous variables were presented as mean and standard deviation, whereas categorical variables as percentage. Unpaired two-tailed t test was used for subgroup comparisons (variable PFI). A *p* value < 0.05 was considered to be significant. In order to calculate the negative predictive value of skin tests to CBDCA, we have analyzed a subgroup of our patient cohort (*n* = 82) who did not previously develop any HSR to CBDCA with negative skin tests to the platinum agent. The positive predictive value of skin tests to CBDCA could not be evaluated because treatments modifications occurred in patients with a positive skin test.

## 3. Results

### 3.1. Study Population and HSRs to Platinum Agents

We evaluated 124 consecutive female patients with a prevalence of ovarian cancer (79%), followed by endometrial cancer (17%) and cervical cancer (4%). During their lifetime, 56% of the patients had received previous CBDCA, 3% had received CDDP, and 41% had received both CBDCA and CDDP. The mean of platinum-based chemotherapy lines was 2.2 (±0.8), with a mean of total cycles of 11.8 (±5.0) and mean time from the last platinum exposure of 18.4 (±15.2) months (Table 1). Among the patients evaluated, 12% were atopic, with a previous history of allergic rhinitis and/or asthma and/or food allergy, 4% had experienced HSRs to NSAIDs, 11% to antibiotics, and 10% to contrast media.

### 3.2. HSRs to Platinum Agents

Among the cohort of patients evaluated, 47% (58/124) had experienced an HSR during previous platinum-based therapy: 35% (44/124) to CBDCA, 5% (6/124) to CDDP, and 7% (8/124) to both platinum agents (Table 1). Most of the HSRs (95%) were classified as immediate and only two (observed following treatment with CBDCA) were classified as delayed (Table 2). Severe anaphylaxis was more frequent with CBDCA than with CDDP (60% vs. 33%) (Table 2).

### 3.3. Role of Skin Tests in the Diagnosis and Prevention of HSRs to Platinum Agents

*In-vivo* skin tests confirmed an IgE-dependent mechanism in 27 out of 42 patients with immediate-HSRs to CBDCA (64% sensitivity) and a T-cell mediated mechanism in one of the two cases of delayed-HSRs to CBDCA (Table 2). Interestingly, among the 42 immediate-HSRs to CBDCA, 15 positive skin tests were identified during the first evaluation and 12 were subsequently detected with re-testing after 3 weeks from the platinum re-challenge. An IgE-dependent mechanism was confirmed in 100% of patients with immediate-HSRs to CDDP and to both CBDCA and CDDP in their lifetime.

In order to assess the negative predictive value (NPV) of skin tests to CBDCA, among the 124 patients evaluated, 82 patients, who did not previously develop any HSR to CBDCA and had negative to the skin tests to the platinum agent, were followed over time. In 66 out of the 82 patients, the negative skin tests accurately predicted the absence of a HSR to CBDCA during re-challenge, yielding a negative predictive value of 80%, whereas 16 out of 82 developed an HSR during CBDCA re-challenge with an anaphylactic reaction (all grade 3) in 11 out of 16.

*In-vivo* skin tests for both culprit and alternative platinum agents, performed in those patients who developed immediate-HSRs, identified a cross-reactivity between the two agents. Eighteen percent of patients with HSRs to CBDCA who had never been treated with CDDP had skin tests positive also to CDDP. Moreover, *in-vivo* skin tests were useful for the detection of sensitized patients. The skin tests demonstrated a sensitization to a platinum agent in 8 of 66 (12%) patients who had never developed an HSR during platinum-based therapy: 6 to CBDCA, 1 to CDDP, and 1 to both CBDCA and CDDP. Interestingly, the positivity was detected during re-testing after 3 weeks from the platinum re-challenge.

According to the cycles and line therapies, the HSRs to CBDCA developed mainly from the 7th cycle of the platinum agent (Figure 1A) and during the 2nd and 3rd line of therapy (Figure 1B).

In the 47 patients analyzed for *BRCA1/2* status, a deleterious mutation was demonstrated in 44.7% (21/47) of the cases. A higher rate of *BRCA1/2* mutations was found in patients with platinum-induced HSRs than in those without HSRs (58.3% vs. 30.4%, *p* = 0.054). Since most of the platinum-induced HSRs were due to CBDCA, we analyzed potential differences between patients with and without HSRs to CBDCA. As reported in Figure 2, we observed a PFI significantly longer in patients who developed immediate-HSRs to CBDCA compared to those who did not (*p* = 0.0039).

### 3.4. Therapeutic Approaches in Patients with HSR to CBDCA with Positive Skin Tests

Among the 27 patients with an immediate-HSR to CBDCA that resulted in positive skin tests to CBDCA and negative to CDDP, 15 were switched to therapy with CDDP without developing any HSR; 5 continued to be treated with CBDCA under desensitization procedure, according to the literature [23,24,25]. Among them, 2 developed an HSR during treatment with a grade of reaction less severe than the previous one. Therefore, they were able to successfully complete the treatment with CBDCA following desensitization protocols. Four out of 27 patients were switched to other drugs and 3 patients referred to other hospitals (Figure 3).

## 4. Discussion

HSRs, including anaphylaxis, to platinum drugs are increasing in frequency due to the drugs’ extensive use in the treatment of gynecological cancers and to repeated exposures in patients with increased life expectancy. Therefore, previous HSRs to platinum drugs limit the use of platinum re-treatment in patients having platinum-sensitive disease, affecting their survival. A systematic allergy work-up might identify an IgE-dependent mechanism underlying immediate-HSRs or a T-cell mediated mechanism in a delayed-HSRs, leading to a more personalized therapy. In fact, desensitization protocols based on the degree of HSRs have allowed patients with previous HSRs to continue a platinum-based therapy, reducing the risk of developing severe anaphylaxis in the following treatments [12,23,24]. Up to now, skin tests have been a fundamental tool for the diagnosis of immediate- as well as delayed-HSRs to platinum agents [11,13,14,15].

In our study we evaluated 124 women with gynecological cancer, predominantly ovarian cancer, who received platinum-based therapy during their lifetime. Among them, we observed 47% of patients who developed HSRs to at least one platinum agent with a prevalence of immediate HSRs. Furthermore, the *in-vivo* skin tests performed during the allergy work-up were able to identify an IgE-dependent mechanism in 67% of patients with a previous CBDCA-induced immediate-HSR and in all the patients who experienced HSRs to CDDP. *In-vivo* skin tests were also useful for the detection of T-cell mediated mechanisms underlying delayed HSRs, performing delayed skin test readings even after 24, 48, 72, and 96 h, based on the timing of the HSRs.

Moreover, we have to take in account that even if the sensitivity of skin tests is approximately 80%, this may be less than 36% when the test is performed after more than 6 months from the reaction [14,18,19]. This issue must always be considered when performing skin tests, since negative results could be due to the long period between the HSR and the performance of the skin tests. Therefore, in these cases, to reduce the risk of false negative results, we performed re-testing after 3 weeks from the re-exposure, doubling the number of positive results. Our results are in line with those obtained by Patil et al. in a cohort of patients with a recent clinical history of HSRs to carboplatin and an initial negative skin tests [18]. Identifying an IgE-mediated mechanism in patients with immediate-HSR to platinum agents is critical for setting-up an alternative therapy such as switching to another platinum agent or adopting a desensitization protocol for CBDCA or CDDP in order to continue a platinum-based therapy. In fact, in patients who experienced an immediate or delayed HSR to CBDCA, an alternative platinum agent such as CDDP is a feasible approach, in order to maintain the beneficial effect of the platinum agents, reducing the risk of HSRs [6,25,26]. However, as the possibility of cross-reactivity between these agents has been reported, a subsequent treatment with CDDP does not exclude future HSRs [17]. Therefore, in order to identify a possible cross-reactivity between platinum agents, we propose to perform *in-vivo* skin tests for both culprit and alternative platinum agents. In our experience, this approach identified a cross-reactivity between these platinum agents in 18% of cases and helped to identify an alternative therapeutic approach. Consequently, if the therapy with platinum is strongly indicated despite immediate-HSRs, the following treatment will be performed administering the platinum agent, following recent protocols of desensitization [23,27,28,29,30].

Because the risk of developing HSRs to platinum agent increases with the number of cycles of therapy [4], it could be safer for the patient to undergo the *in-vivo* skin tests before re-exposure to the platinum agent, especially in those who have to undergo a 2nd or 3rd line therapy for recurrent disease. In our experience, this approach identified 12% of patients who did not experience any HSRs but that became sensitized during the previous platinum-based therapy. Interestingly, the positive results were obtained during re-testing, supporting the importance of this approach during the allergy work-up. Therefore, if more than 6 months have passed since the last platinum-based therapy, *in-vivo* skin tests must be repeated after 3 weeks from re-exposure, in order to reduce the risk of false negative results.

## 5. Conclusions

On the basis of our findings, a skin test for platinum agents can be considered a simple and sensitive tool for the diagnosis of both immediate- and delayed-HSRs to CBDCA and/or CDDP, and can help to identify any sensitization to platinum agents, preventing future reactions. Moreover, performing in parallel skin tests for both culprit and alternative platinum agents can be useful for detecting possible cross-reactivity. In conclusion, the allergy work-up employed in our cohort of patients represents a comprehensive approach in patients under platinum-based therapy, contributing to design of a more personalize treatment.

## Figures and Tables

**Figure 1 cancers-13-05468-f001:**
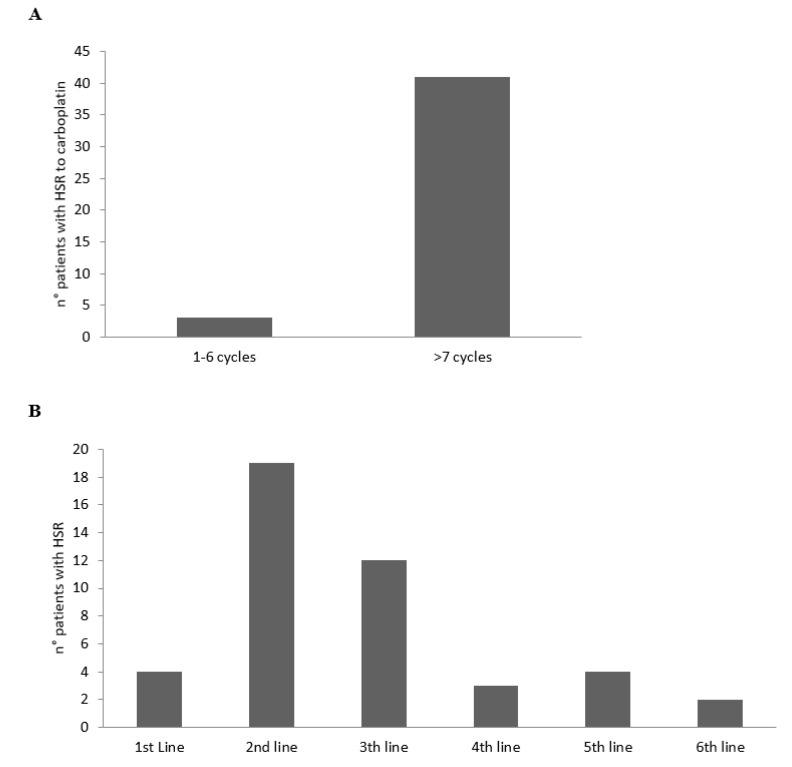
Number of patients with HSRs to CBDCA according to the cycles (**A**) and lines (**B**) of therapy.

**Figure 2 cancers-13-05468-f002:**
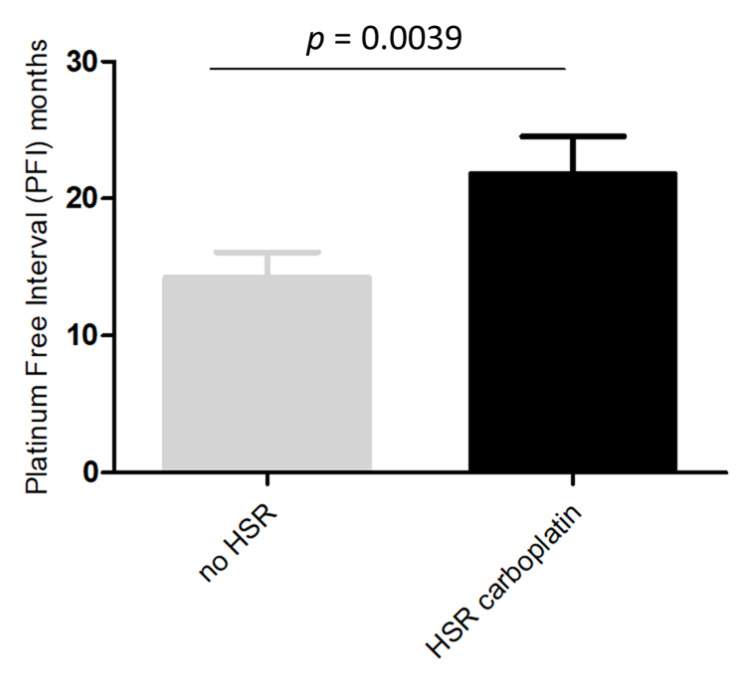
Platinum free interval in patients with and without immediate-HSRs to CBDCA.

**Figure 3 cancers-13-05468-f003:**
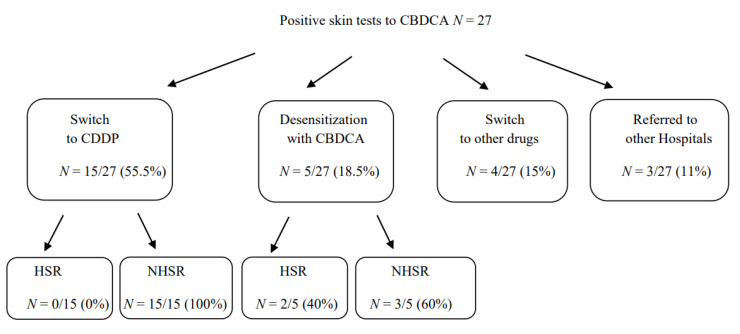
Therapeutic approaches in patients with HSR to CBDCA with positive skin tests. Abbreviations: HSR, hypersensitivity reactions.

**Table 1 cancers-13-05468-t001:** Patients′ characteristics.

*N* of Patients	124
Age (mean ± SD)	61.9 ± 10.6
Ovarian cancer *n* (%)	98 (79)
Endometrial cancer *n* (%)	21 (17)
Cervical cancer *n* (%)	5 (4)
Prior CBDCA therapy *n* (%)	70 (56)
Prior CDDP therapy *n* (%)	3 (3)
Prior CBDCA and CDDP *n* (%)	51 (41)
Lines of treatment (mean ± SD)	2.2 ± 0.8
Cycles of treatment (mean ± SD)	11.8 ± 5.0
Time from the last platinum exposure (mean ± SD)	18.4 ± 15.2
Atopy *n* (%)	15 (12)
HSR to NSAIDs *n* (%)	5 (4)
HSR to antibiotics *n* (%)	11 (11)
HSR to contrast media *n* (%)	13 (10)
HSR to platinum agents *n* (%)	58 (47)
HSR to CBDCA *n* (%)	44 (35)
HSR to CDDP *n* (%)	6 (5)
HSR to CBDCA and CDDP *n* (%)	8 (7)

Abbreviations: SD, standard deviation; HSR, hypersensitivity reactions; NSAIDs, non-steroidal anti-inflammatory drugs.

**Table 2 cancers-13-05468-t002:** Characteristics of HSRs to platinum agents and skin test results.

Variation	CBDCA	CDDP	CBDCA/CDDP
Immediate *n*°/total HSR (%)	42/44 (95)	6/6 (100)	8/8 (100)
-Positive skin tests *n*°/total immediate-HSR (%)	27/42 (64)	6/6 (100)	8/8 (100)
-Negative skin tests *n*°/total immediate-HSR (%)	15/42 (36)	0/6 (0)	0/8 (0)
Not anaphylaxis (grade 1 and 2) *n*/total immediate-HSR (%)	17/42 (40)	4/6 (67)	3/8 (37)
Anaphylaxis (grades 3–5) *n*/total immediate-HSRs (%)	25/42 (60)	2/6 (33)	5/8 (63)
Delayed *n*°/total HSR (%)	2/44 (5)	0 (0)	0 (0)
-Positive skin tests *n*/total delayed-HSR (%)	1/2 (50)	-	-
-Negative skin tests *n*/total delayed-HSR (%)	1/2 (50)	-	-

Abbreviations: HSRs, hypersensitivity reactions; SPT, skin prick tests; IDT, intradermal tests.

## Data Availability

The data presented in this study are available on request from the corresponding author. The data are not publicly available due to privacy.

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
