# Peer review of "The Role of Skin Tests in the Prevention and Diagnosis of Hypersensitivity Reactions to Platinum Agents in Gynecological Cancer: A Single-Center Italian Retrospective Study"

_cancers, 2021, doi:10.3390/cancers13215468_

Round 1
Reviewer 1 Report
Dear authors, the paper, to me, is now suitable for publication
Reviewer 2 Report
Title
Succinct and concise In the current version
Introduction
Intro was well written.
Statistical analysis
A screening/diagnositc test was performed, such as sensitivity, specificity, negative predictive value etc. However, the patient number (N=82) for the assessment of negative predictive value is doubtful.
Results
The sensitivity of in-vivo skin test to detect immediate HSR to CBDCA was 64%, lower than 80% in the literature. Authors overstated that skin test was a sensitive tool in this investigation for the diagnosis and prevention of HSRs to CBDCA and/or CDDP and can be useful for detecting possible cross-reactivity among platinum agents.
According to data in Tables, patients who did not previously develop any HSR to CBDCA [N=72 (124-44-8)] and were resulted negative to the skin tests to the platinum agent, were followed over time. The patient number (N=82 used) for the assessment of negative predictive value is not relevant.
Table 2.
Data was still not well arranged to each item.
Figure 2.
The legend is now correctly corresponding to Figure 2.
Comments:
The data presented in this revision version could not convince me to recommend its publication due to the limited accuracy and disputed calculation.
This manuscript is a resubmission of an earlier submission. The following is a list of the peer review reports and author responses from that submission.
Round 1
Reviewer 1 Report
With interest I have read this manuscript which is overall well written, well structured and clear.
In the present article the authors analyzed the frequency of both type I and Type IV hypersensitivity reactions in patients with gynecological cancer treated with carboplatin (CBDCA) and/or cisplatin (CDDP). They have deeply investigated the role of performing in-vivo skin tests in the diagnosis and prevention of HSRs.
Since the probability of a hypersensitivity reaction to platinum compound increases with increasing cycles of therapy, a well performed and organized allergy work-up for the diagnosis and prevention of HSRs is required in managing patients with gynecological cancers. Unfortunately, not all clinical centers carry out allergy evaluations in the context of chemotherapy. And the experience of this center could be helpful for other clinicians.
The allergy approach of this Italian center can be taken as an example, and It can be reproduced in different realities. The allergy work-up reported in this original article adds essential elements not only in the diagnostic approach of HSRs, but also in the prevention of these reactions. The allergy work-up performed before starting new cycle of platinum and re-evaluating the patient with a retesting after 3 weeks from the therapy could be a useful approach. This element is little treated in literature and allows to detect sensitization to the platinum agent that would not have been highlighted without allergy evaluation.
The methodological diagnostic approach is rigorous as shown in the paragraph on materials and methods and in the results. Some diagnostic tools used in the study are well documented in the literature.
The conclusions of the authors are consistent with the results obtained. It would be interesting to subsequently evaluate these patients following therapeutic changes according to the results obtained by the allergy work-up.
The figures and tables are clear and exhaustive, but the left alignment of the text body in the table would be easier to read. Moreover, I would add a single table or diagram which summarizes the methodology used.
Specific comments:
- on Table 1 and 2 I would recommend, in order to easily read the results, the flush left alignment of the body of the text. Centered text is less readable.
- on Fig. 2 there is a typo in the word carboplatin
Reviewer 2 Report
The authors aimed to prevent and diagnose HSR to platinum agents by skin tests. However, this manuscript can not conclude that skin test is useful to prevent and diagnose HSR due to insufficient data. We also know that skin test is old-fashioned and no longer used to predict HSR to antibiotics as well.
Reviewer 3 Report
General
Summary
Objective: To analyze the frequency of HSRs in patients with gynecological cancer treated with CBDCA and/or CDDP and to evaluate the role of skin tests in the diagnosis and prevention of HSRs.
Design: Retrospective cohort
Setting: One academic hospital
Population: 124 patients with salvage platinum-based treatment for recurrent gynecological cancer after a prior treatment withCBDCA (56%), CDDP/ CBDCA (41%), or CDDP (3%) at the Azienda Ospedaliera Universitaria Pisana (AOUP) between 2011 and 2018
Methods: Skin prick test (SPT) and Intraderamal test (IDT) was conducted by specific schedule.
Main Outcome Measures: Frequencies of HSRs to platinum agents. The ratios of positive and negative results in total HSRs, total immediate HSRs, and total delayed HSRs. Number of patients with HSRs to CBDCA according to the cycles and lines of therapy. PFI between patients with immediate HSRs to CBDCA and those without.
Results: Skin tests confirmed an IgE-dependent mechanism in 67% of patients with immediate-HSRs to CBDCA and allowed to identify a cross-reactivity between platinum agents in 18% of patients. Moreover, among those who had never developed an HSRs during platinum-based therapy, in-vivo skin tests allowed to identify 12% of sensitized patients. Authors suggested skin test was a sensitive tool for the diagnosis and prevention of HSRs to CBDCA and/or CDDP and can be useful for detecting possible cross-reactivity among platinum agents.
Title
Succinct and concise
Introduction
Intro was well written.
Methods
Skin prick test (SPT) was used prior to re-exposure of platinum, not before each platinum-chemotherapy course. Intraderamal test (IDT) was performed 3 weeks after re-exposure of platinum if the SPT result showed negative. Unpaired two-tailed t test was used for subgroups comparisons (variable PFI).
Statistical analysis
Any control group was not included. No accuracy of a screening/diagnositc test was performed, such as sensitivity, specificity, negative predictive value etc.
Results
Authors focused on positive results. Most data is presented descriptively without a proposed confidence interval. Since there is a lack of analyses for accuracy of a screening/ diagnostic test, authors overstated that skin test was a sensitive tool for the diagnosis and prevention of HSRs to CBDCA and/or CDDP and can be useful for detecting possible cross-reactivity among platinum agents.
Table 2. Data was not well arranged to each item.
Figure 2. The legend is not correctly corresponding to Figure 2.
Comments:
I look forward to the improvements in data analyses and presentation, consistent with the title and study purposes.